# 🤝 VLIS: Unimodal Language Models Guide Multimodal Language Generation

**Jiwan Chung** and **Youngjae Yu**
Yonsei University

https://github.com/JiwanChung/vlis

## Abstract

Multimodal language generation, which leverages the synergy of language and vision, is a rapidly expanding field. However, existing vision-language models face challenges in tasks that require complex linguistic understanding. To address this issue, we introduce Visual-Language models as Importance Sampling weights (🤝 VLIS), a novel framework that combines the visual conditioning capability of vision-language models with the language understanding of unimodal text-only language models without further training. It extracts pointwise mutual information of each image and text from a visual-language model and uses the value as an importance sampling weight to adjust the token likelihood from a text-only model. VLIS improves vision-language models on diverse tasks, including commonsense understanding (WHOOPS, OK-VQA, and ScienceQA) and complex text generation (Concadia, Image Paragraph Captioning, and ROCStories). Our results suggest that VLIS represents a promising new direction for multimodal language generation.

## 1 Introduction

Visual Language Models (VLMs) extend unimodal text-only language models by conditioning their outputs on image context. Recent VLMs (Li et al., 2022a, 2023b; Wang et al., 2022) can perform diverse multimodal tasks from commonsense VQAs (Marino et al., 2019; Schwenk et al., 2022) to in-context learning (Alayrac et al., 2022; Awadalla et al., 2023; Huang et al., 2023). Moreover, instruction tuning with visual inputs (Liu et al., 2023; Li et al., 2023a; Dai et al., 2023) has improved the VLMs' responsiveness to an even more extensive variety of tasks (Lu et al., 2022a; Yang et al., 2021).

However, most VLMs only partially inherit the linguistic understanding capability of the unimodal models (Iki and Aizawa, 2021). We here illustrate two intriguing failure cases of the recent VLMs,

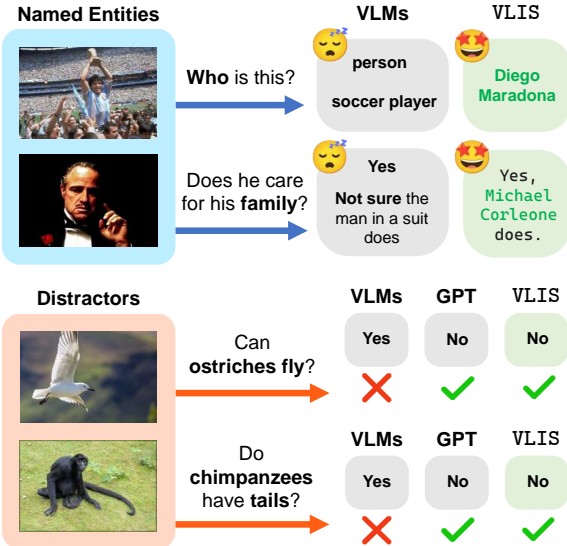

Figure 1: **TOP**: VLIS correctly recognizes named entities, unlike the VLMs. **Bottom**: VLIS is not deceived by the distractor images. Note that the images show a seagull and a monkey, not an ostrich and a chimpanzee. VLIS inherits this favorable linguistic capability from a text-only language model (Touvron et al., 2023; Zhang et al., 2022), and use VLMs as a guide for visual alignment. The examples are truncated for visualization purposes: we provide the full-text in appendix A.2.

using both a strong image captioning model (BLIP-2 (Li et al., 2023b)) and an instruction-tuned model (LLAVA (Liu et al., 2023)). Firstly, VLMs avoid specifying named entities. The upper examples of Figure 1 show the VLM failing to describe a public figure (*Diego Maradona*) or movie character (*Don Corleone*). The problem is not the lack of knowledge: after applying our zero-shot method (VLIS), the VLM tells the names. We further investigate this phenomenon in the landmark recognition experiment in appendix A.1.

Secondly, VLMs rely on the image context, even when they should not. The lower examples of the same figure show the VLM being misled by image context to deny commonsense knowledge. The

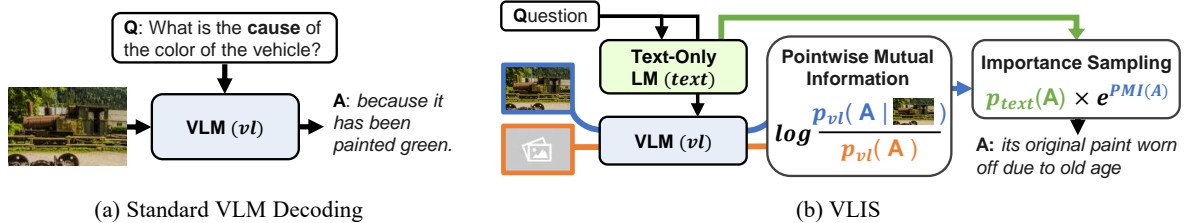

(a) Standard VLM Decoding                (b) VLIS

Figure 2: Comparison of VLIS and standard VLM decoding process. Using the VLM, we first obtain the image-conditional $p_{vl}(Answer|image)$ and text-only likelihood $p_{vl}(Answer)$ given an image and a prompt or question. Then, we compute the exponentiated pointwise mutual information (PMI) with the likelihoods. Finally, the exponentiated PMI score is used as the importance weights for the text-only model likelihood $p_{text}(Answer)$.

questions are not unanswerable: the text-only language model without the image context answers both correctly. We provide more samples on visual distraction in appendix A.2.

Hence, the linguistic capabilities of the VLMs are not optimal yet. On the other hand, the unimodal text-only language models themselves (Brown et al., 2020; Touvron et al., 2023) show reliable linguistic understanding and known for their knowledge understanding (Petroni et al., 2019; Meng et al., 2022) and complex reasoning capabilities (Kojima et al., 2022; Qin et al., 2023). Hence, it becomes reasonable to delegate the burden of language modeling to the text-only models.

To this end, we propose Visual-Language models as Importance Sampling weights (🤝 VLIS) as a plug-and-play method to enhance the unreliable linguistic understanding of the VLMs. When generating each text token, VLIS follows the token likelihoods of the unimodal **text-only** language model. Furthermore, VLIS multiplies importance sampling (Tokdar and Kass, 2010) weights derived from a VLM to provide the **visual alignment** signals. To isolate the visual conditioning capability of the VLMs from their language modeling preference, we incorporate the exponentiated pointwise mutual information (PMI) (Church and Hanks, 1990) of the image context and the current text token as the weights. As a result, VLIS can maintain the favorable language modeling capability of the text-only model and control the visual conditioning strength simultaneously.

We evaluate VLIS on two VLM backbones to test whether VLIS is effective both when the language modeling capability of the VLM is weaker than that of the text-only model (BLIP-2 (Li et al., 2023b)) and when the VLM is expected to model language well owing to the visual instruction tuning process (LLAVA (Liu et al., 2023)). Our ex-

periments consist of various tasks that require both reliable language modeling and strong visual conditioning, including weirdness identification (WHOOPS (Bitton-Guetta et al., 2023)) and commonsense VQA (OK-VQA (Marino et al., 2019), ScienceQA (Lu et al., 2022a)), extended image captioning (Concadia (Kreiss et al., 2022) and Image Paragraph Captioning (Krause et al., 2017)), and open-ended generation (ROCStories (Mostafazadeh et al., 2016)). Compared to the dataset-specific state-of-the-art baselines and the base VLMs, VLIS improves linguistic capabilities such as responsiveness to prompts while maintaining visual conditioning according to a comprehensive set of evaluation metrics.

## 2 VLMs as Importance Sampling Weights

We propose Visual-Language models as Importance Sampling weights (VLIS) to harmonize the visual conditioning capability of the VLMs with the linguistic fluency of the text-only language models. We provide the intuition behind our approach in §2.1, describe our token-level visual alignment scores in §2.2, and combine the said scores with the text-only model via importance sampling in §2.3.

### 2.1 Intuition

Many recent Visual Language Models (VLMs) (Li et al., 2023b; Alayrac et al., 2022; Liu et al., 2023) are often built on top of text-only language models (Iyer et al., 2022; Hoffmann et al., 2022; Touvron et al., 2023). At each timestep $t$, the per-token likelihood of the autoregressive text-only language models is modeled as $p_{text}(x_t|x_{<t})$, where $x$ denotes a text token. To build a VLM $p_{vl}$, one can finetune the text-only model on data $S$ consisting of paired image $c$ and text $x$ with maximum likelihood estimation as the objective.

$$\theta_{vl} \sim argmin_\theta E_{(x,c)\in S}[-\log p_\theta(x|c)] \quad (1)$$

However, while this objective only maximizes the image-conditional likelihood $p_{vl}(x_t|c)$, it may lead to unpredicted artifacts in the marginal likelihood $p_{vl}(x_t)$ that does not depend on any particular image. For example, image captioning models are known to not only reflect but also amplify the social bias present in the training data (Hendricks et al., 2018), or distort the original language model's commonsense knowledge as described in §1.

We henceforth seek to extract the visual conditioning capability of the VLMs isolated from their dubious language modeling skills.

## 2.2 Extracting Visual Weights

Here, we aim to find a quantity that extracts the visual conditioning strength of a VLM stripped of its language modeling preference. We employ Pointwise Mutual Information (PMI) (Church and Hanks, 1990), which measures the association between two events (text and image in our setting). On each step, we want to compute the PMI between the image context $c$ and the next token $x_t$ given the previous text context $x_{<t}$:

$$PMI(x_t|c, x_{<t}) = \log \frac{p_{vl}(x_t, c|x_{<t})}{p_{vl}(x_t|x_{<t})p_{vl}(c)} \quad (2)$$

$$= \log \frac{p_{vl}(x_t|c, x_{<t})}{p_{vl}(x_t|x_{<t})} \quad (3)$$

eq. (3) reformulates the definition in eq. (2) for better tractability. The numerator is the image-conditional likelihood of the VLM and is easily obtainable. However, the denominator requires marginalization over the image context $c$. We enumerate three proxies below that bypass the excessive computation required to obtain the expectation over all possible images.

**Approximating the marginal**. The first approximation is training a separate text-only model with the VLMs' training data $S$. Considering the massive scale of dataset $S$, this option requires a considerable burden of additional training. Also, there is no guarantee that the newly trained model will accurately estimate the marginal likelihood due to the additional complexity of training another model. The second option is using a sample mean of the pre-selected image set as a proxy to the real mean. Lastly, the score for only one or two images might suffice as the sample image set.

We use the last method with the least computational overhead. Here, the sample set is a tiny set of images with close to no visual information. In practice, we use two images: a black-filled image $c_b$ and a white-filled image $c_w$.

$$p_{vl}(x_t|x_{<t}) \sim \frac{1}{2} \sum_{c \in [c_b, c_w]} p_{vl}(x_t|x_{<t}, c) \quad (4)$$

This efficient alternative works reasonably well in practice and is used in all our experiments. As a result, VLIS runs three forward passes of VLM (one for the conditional likelihood and two for the marginal likelihood) and a single pass of the text-only model on each step of the generation process. We explore more choices of selecting the marginal image set later in appendix C, which shows that our specific set of images provides a reasonable trade-off between generation quality and inference time.

## 2.3 Computing VLIS Scores

We start from the token likelihood of text-only language models $p_{text}(x_t|c, x_{<t})$. To control the degree of confidence in the text-only models' decisions, we introduce a language temperature $\tau$ to smooth or de-smooth the text-only distributions:

$$\bar{p}_{text}(x_t|c, x_{<t}) \propto p_{text}(x_t|c, x_{<t})^{\frac{1}{\tau}} \quad (5)$$

Then, we multiply the exponentiated PMI introduced in §2.2 with the likelihood for better visual alignment. VLIS decides the next token $x_t$ with the following score $f(x_t)$:

$$f(x_t) = \bar{p}_{text}(x_t|c, x_{<t})e^{PMI(x_t, c|x_{<t}))} \quad (6)$$

$$= \bar{p}_{text}(x_t|c, x_{<t})\frac{p_{vl}(x_t|c, x_{<t})}{p_{vl}(x_t|x_{<t})} \quad (7)$$

Written as eq. (7), VLIS performs importance sampling of the smoothed text-only model likelihood $p_{text}$. Importance sampling (Tokdar and Kass, 2010) is a Monte-Carlo method of estimating a quantity $v(x)$ from the *nominal distribution* $p(x)$ with samples from another distribution called *importance distribution* $q(x)$. The estimated quantity here is the text-only model likelihood $\bar{p}_{text}(x_t)$, the nominal distribution is the VLMs' image-conditioned likelihood $p_{vl}(x_t|c)$, and the importance distribution is the marginal $p_{vl}(x_t)$.

$$E[f(x_t) : P] \sim E_{x_t \sim q(x_t)}[v(x_t)\frac{p(x_t)}{q(x_t)}] \quad (8)$$

$$v(x_t) := \bar{p}_{text}(x_t|x_{<t})$$
$$p(x_t) := p_{vl}(x_t|c, x_{<t})$$
$$q(x_t) := p_{vl}(x_t|x_{<t})$$

Implementation-wise, we replace the expectation with a single sample (current generated text). Thus, VLIS effectively treats the current token candidate as sampled from the VLMs' marginal $p_{vl}(x_t)$ and reweigh its importance with the VLMs' conditional $p_{vl}(x_t|c)$.

**Fluency masking.** The log visual weights $PMI(x_t, c|x_{<t})$ of VLIS is a log-likelihood ratio and is unbounded. Hence, some extreme cases, such as tiny values of the marginal likelihood $p_{vl}(x_t|x_{<t})$ may overrule the language generation process of the text-only model, yielding degenerate text. To prevent such text degeneracy, we apply a fluency mask to our importance sampling score $f(x_t|x_{<t}, c)$: only the tokens with text-only likelihood larger than the threshold $\alpha$ are allowed to be selected. We omit the dependency on the contexts $x_{<t}, c$ in the equation below for simplicity.

$$\tilde{f}(x_t) = \begin{cases} f(x_t), & if \ x_t \in \mathcal{V}_{top} \\ -inf, & otherwise \end{cases} \quad (9)$$

$$\mathcal{V}_{top} = \{x_t | p_{text}(x_t) \geq \alpha\} \quad (10)$$

Intuitively, this mask filters out any token candidates the text-only model sees as the next token with a probability lower than $\alpha$. We fix the fluency threshold to $\alpha = 0.001$ in all experiments except for an alternative architecture (appendix E). Still, VLIS is not overly sensitive to the specific value of the fluency threshold. We conduct a hyperparameter search experiment to verify this in appendix D.

The token that maximizes this final score $\tilde{f}(x_t|c, x_{<t})$ is greedily selected as the next token. When VLIS is combined with other decoding methods, such as beam search, the score substitutes the original token likelihood as per-token scores.

## 3 Experiments: Describing Facts

We verify that VLIS can alleviate the factual inaccuracy concern raised in Figure 1 with various multimodal benchmarks: weirdness identification §3.1, commonsense understanding §3.2, and scientific reasoning §3.3. VLIS consistently outperforms the backbone VLM and shows comparable factual correctness to the strong baselines.

**Experimental setups.** We explore two experimental setups. Our experiments on the WHOOPS dataset incorporate LLAVA (Liu et al., 2023) and Lynx (Zeng et al., 2023) as the VLMs and Vicuna 7B (Chiang et al., 2023) as the text-only model. In the VQA experiments, we use BLIP-2 OPT 2.7B (Li et al., 2023b) and OPT IML Max

| Models | Pipe | 0-shot | Acc (%) |
|---|---|---|---|
| Chance | | | 50 |
| BLIP-2 | | ✓ | 50 |
| BLIP-2 | | | 73 |
| Model Caption | ✓ | ✓ | 59 |
| GT Caption | ✓ | ✓ | 74 |
| VLM (LLAVA) | ✓ | ✓ | 59 |
| VLM (Lynx) | ✓ | ✓ | 71 |
| Ours (LLAVA) | ✓ | ✓ | 73 |
| Ours (Lynx) | ✓ | ✓ | **80** |

Table 1: Results in the *identification of weird images* task of WHOOPS dataset (Bitton-Guetta et al., 2023). *Pipe* represents further pipelining with GPT3 and *0-shot* denotes a zero-shot method. The best numbers are **bolded** and the second best ones are underlined.

1.3B (Iyer et al., 2022) as our backbones.[1] Note that the choices of model pairs are intentional: we impose similar computational requirements on both the VLM and the text-only model to limit the additional computational burden of VLIS. In both cases, we use the base VLM as a general baseline to evaluate the gain from VLIS. Also, to verify the contribution of the PMI weights, we implement Naïve Ensemble which simply multiplies the token likelihood of the VLM and the text-only model.

**Evaluation metrics.** We evaluate closed-ended questions with binary (WHOOPS) and multichoice (ScienceQA) accuracy. The open-ended VQAs (OK-VQA and VQAv2) use the task-specific VQA metric (Antol et al., 2015).

### 3.1 Identification of Weird Images

WHOOPS (Bitton-Guetta et al., 2023) is a visual commonsense benchmark to check a VLM's capability to understand images that defy commonsense. We adopt *identification of weird images*, a subtask of the WHOOPS benchmark, which tasks a model to discriminate potentially weird images.

**Approach and Baselines.** Following the original paper (Bitton-Guetta et al., 2023), we employ *pipelining* to turn the original binary classification problem into a description generation problem. Specifically, *pipelining* means that a model first generates explanation-of-violation (EoV) description of the given two images, which is then

---

[1]We assign different tasks for different backbones for fair comparisons: BLIP-2 fails to generate long explanation-of-violation since it is only trained on short captions, while it is not trivial to evaluate LLAVA on short-answer VQAs in a zero-shot manner due to its tendency to generate long explanations.

**What is unusual about this image?**

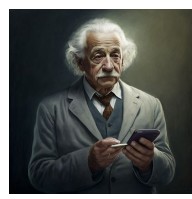

**VLIS**
**Albert Einstein** holding a smartphone is unusual because he lived before the invention of such technology.

**VLM**
it is not common to see a **professional** in a field like science using a cell phone.

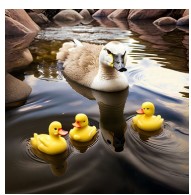

**VLIS**
Duck swimming alongside baby **rubber ducks** in the image is unusual as they are not a real animal.

**VLM**
It is not very common to see a large bird with **three ducklings** at once.

Figure 3: Qualitative samples from WHOOPS (Bitton-Guetta et al., 2023) experiments. As marked in **green**, specific descriptions are required to explain weirdness.

processed to the off-the-shelf text-only classifier GPT3 (Brown et al., 2020) to yield a binary decision on which image is weird. We use VLIS to generate such EoV descriptions. The pipelined baselines include EoV from the backbone VLM (LLAVA), conventional machine-generated captions, and ground-truth captions from the WHOOPS dataset. We also include pipeline-less BLIP-2 (both supervised and zero-shot) as a baseline. The same prompt we used for both VLIS and the backbone VLM is illustrated in appendix F.

**Results**. Table 1 and Figure 3 presents results with LLAVA (Liu et al., 2023), an instruction-tuned VLM. VLIS-generated weirdness explanations perform on par with the ground-truth captions, which are manually annotated to contain details necessary to identify the strangeness. Also, our method as a zero-shot method shows comparable performance to the supervised baseline BLIP-2. Interestingly, LLAVA alone cannot outperform conventional captions, even with instruction tuning and prompting.

## 3.2 Commonsense Understanding

Unimodal language models embody commonsense knowledge (Petroni et al., 2019; Davison et al., 2019; Tamborrino et al., 2020). If VLIS can inherit this commonsense understanding capability, it would outperform the base VLM in tasks requiring both commonsense and visual understanding. Here, we examine this possibility with a commonsense VQA benchmark of OK-VQA (Marino et al., 2019). Further, VLIS is also shown to maintain visual specificity in VQAv2 (Goyal et al., 2017).

**Approach and baselines**. We use OK-

| Models | V | L | OKVQA | VQAv2 |
|---|---|---|---|---|
| FewVLM | ✓ | | 16.5 | 47.7 |
| Frozen | ✓ | | 5.9 | 29.6 |
| VLKD | ✓ | | 13.3 | 42.6 |
| BLIP-2 | ✓ | | 31.7 | 53.5 |
| OPT-IML | | ✓ | 19.1 | 36.0 |
| Naïve Ensemble | ✓ | ✓ | 26.6 | 34.6 |
| Ours | ✓ | ✓ | **34.2** | **53.6** |

Table 2: Results in the validation set of OK-VQA (Marino et al., 2019) and VQAv2 (Goyal et al., 2017). *V* denotes using a VLM and *L* denotes using a unimodal language model.

| Models | IMG | TXT | NO | ALL |
|---|---|---|---|---|
| UnifiedQA$_{Small}$ | 44.1 | 50.2 | 44.5 | 45.8 |
| UnifiedQA$_{Base}$ | 48.1 | 53.1 | 46.7 | 48.5 |
| GPT-3 | 65.7 | 74.2 | 79.6 | 74.0 |
| BLIP-2 | 35.5 | 34.6 | 24.2 | 28.2 |
| OPT-IML | 45.4 | 52.2 | 49.8 | 49.0 |
| Naïve Ensemble | 45.9 | 53.6 | 49.7 | 49.7 |
| Ours | **49.3** | 53.1 | 49.1 | **50.2** |

Table 3: Zero-shot results on ScienceQA test set (Lu et al., 2022a). *IMG* denotes subset with image context, *TXT* the text context subset, and *NO* the subset without any context.

VQA (Marino et al., 2019) as an example of commonsense-augmented VQA and VQAv2 (Goyal et al., 2017) as a visually intensive VQA problem. We compare VLIS with strong VLM models, including FewVLM (Jin et al., 2022), Frozen (Tsimpoukelli et al., 2021), and VLKD (Dai et al., 2022).

**Results: commonsense knowledge**. In the OK-VQA (Marino et al., 2019) experiment in Table 2, we show that VLIS achieves meaningful development over the backbone VLM (BLIP-2). Also, the text-only backbone (OPT-IML) and Naïve Ensemble perform substantially worse, proving that VLIS is not just imitating the text-only model outputs. Instead, VLIS adaptively fuses the commonsense understanding capability of the text-only model with the visual conditioning of the VLM.

**Results: maintaining visual specificity**. When VQAs do not require text-based reasoning, VLIS should focus on visual conditioning only. The rightmost column of Table 2 summarizes results on VQAv2 (Goyal et al., 2017) dataset, a VQA dataset that has its textual bias intentionally removed. As

| Model | Zeroshot | Cap | Desc |
|---|:---:|---|---|
| Kreiss et al. | | 11.3 | 17.4 |
| Socratic Model | ✓ | 38.9 | 22.6 |
| BLIP-2 | ✓ | 20.0 | **30.6** |
| Naïve Ensemble | ✓ | 24.7 | 18.4 |
| Ours | ✓ | **44.1** | 28.3 |

Table 4: Results on Concadia (Kreiss et al., 2022) test set. *Cap* denotes caption and *Desc* description annotations. We report CIDEr following the literature.

| Model | Shots | M | C | B4 |
|---|---|---|---|---|
| Krause et al. | Full | 16.0 | 13.5 | 8.7 |
| Liang et al. | Full | 17.1 | 16.8 | 9.0 |
| SCST | Full | 13.6 | 13.8 | 5.9 |
| SCST$_{Rep.\ Penalty}$ | Full | 17.9 | 30.6 | 10.6 |
| HSGED | Full | 18.3 | 36.0 | 11.3 |
| PaG-MEG-SCST | Full | 18.2 | 29.4 | 11.5 |
| BLIP-2 | 3 | 10.8 | 6.5 | 4.9 |
| OPT-IML | 3 | 9.5 | 2.5 | 2.2 |
| Naïve Ensemble | 3 | 9.8 | 6.0 | 3.6 |
| Ours | 3 | **14.6** | **14.8** | **6.4** |

Table 5: Results on the Paragraph Captioning (Krause et al., 2017) test set. *M* denotes METEOR, *C* CIDEr, and *B4* Bleu-4 scores.

shown in the VQA score, VLIS (Ours) preserves the VQA capability of the backbone VLM (BLIP-2). Note that Naïve Ensemble falls behind the text-only backbone (OPT-IML), offering a poor trade-off between visual and linguistic understanding.

### 3.3 Scientific Reasoning

ScienceQA (Lu et al., 2022a) evaluates multimodal science reasoning capability. Here, the goal of VLIS would be to improve the answers in the presence of image contexts (IMG) and preserve the answers from the text-only model in the absence of such visual context (TXT and NO).

**Baselines**. We compare our zero-shot VLIS against zero-shot baselines including a VLM (UnifiedQA (Khashabi et al., 2020)) and a text-only language model (GPT-3 (Brown et al., 2020)).

**Results**. Table 3 demonstrates the findings in ScienceQA. On IMG split, VLIS significantly improves the text-only OPT-IML and Naïve Ensemble baselines. Also, VLIS maintains the performance of the text-only backbone in TXT and NO split. Finally, the base VLM (BLIP-2) falls behind by a wide margin, indicating that solid language understanding is necessary for scientific reasoning.

## 4 Experiments: Text Generation

In addition to factual knowledge, text-only language models manifest two critical capabilities: following prompt instructions and generating fluent and diverse text. We demonstrate that VLIS extends these qualities to the visual domain with contextualized captioning (§4.1), paragraph captioning (§4.2), and visual story generation (§4.3).

**Metrics**. Both captioning benchmarks use automatic text metrics, including CIDEr (Vedantam et al., 2015), METEOR (Banerjee and Lavie, 2005), and Bleu-4 (Papineni et al., 2002). In the open-ended generation problem of visual storytelling, we use various fluency metrics (2-gram repetition, diversity, coherence, MAUVE (Pillutla et al., 2021)) and a visual strength metric (CLIPScore (Hessel et al., 2021)). Refer to (Su et al., 2022a) for details on the fluency metrics.

### 4.1 Contextualized Captioning

Concadia (Kreiss et al., 2022) is an image captioning dataset with the additional context of a paragraph from the Wikipedia article. The dataset provides two types of annotations: *caption*, which takes the article into account and *description*, which ignores the article context.

**Approach and Baselines**. Following the original evaluation scheme (Kreiss et al., 2022), we generate a single text to compare against both the ground-truth *caption* and *description*. We include both supervised (Kreiss et al., 2022) and zero-shot (Socratic Model (Zeng et al., 2022)) baselines.

**Result**. In Table 4, VLIS outperforms the Socratic Model (Zeng et al., 2022) implementation based on a stronger language model (GPT-3 175B (Brown et al., 2020)). Interestingly, the base VLM (BLIP-2) and VLIS (Ours) show a completely different text style. VLIS captions are better aligned with *caption*-style, showing that our method reflects the Wikipedia article better than the baselines. On the other hand, the VLM generates *description*-style texts better. Still, VLIS captions are similar to the visually intensive caption (*description*) compared to all other baselines except for the VLM.

### 4.2 Paragraph Captioning

Image Paragraph Captioning (Krause et al., 2017) has paragraph-long captions that describe the image in finer detail than sentence-level captions.

**Approach and baselines**. We saw that neither the VLM nor the text-only model could follow the

| Models | rep-2↓ | div.↑ | coh.↑ | Mauve↑ | CLIP.↑ |
|---|---|---|---|---|---|
| Cont. Search | 2.60 | 0.97 | 0.34 | 0.86 | 0.65 |
| MAGIC | 2.49 | 0.97 | 0.38 | 0.85 | 0.68 |
| BLIP-2 | 24.26 | 0.39 | 0.32 | 0.47 | **0.87** |
| Naïve Ensemble | **1.85** | **0.98** | 0.27 | 0.93 | 0.67 |
| Ours | 2.31 | 0.97 | **0.38** | **0.96** | 0.72 |

Table 6: Results in the ROCStories story generation dataset (Mostafazadeh et al., 2016). *rep-2* denotes 2-gram repetition, *div.* diversity, *coh.* coherence, and *CLIP.* CLIPScore. Higher is better except for *rep-2*.

style of the ground-truth annotation in early experiments. Hence, we provide the model with three in-context examples (3-shot). Note that the setting is still much more challenging compared to that of the fully supervised baselines (Krause at el. (Krause et al., 2017), Liang et al. (Liang et al., 2017), SCST with repetition penalty (Melas-Kyriazi et al., 2018), HSGED (Yang et al., 2020), and PaG-MEG-SCST (Nguyen and Fernando, 2022)).

**Results**. As visible in Table 5, VLIS greatly improves the base VLM (BLIP-2) to generate paragraph captions comparable to the supervised baselines. We provide an interpretation of this improvement in qualitative samples in appendix G: VLIS shows less text degeneracy than the base VLM, while keeping visual hallucination at a minimum unlike Naïve Ensemble.

### 4.3 Story Generation

Story generation is an open-ended generation task. To excel at it, VLIS should generate open-ended text without falling into text degeneracy, all the while staying close to the image context.

**Approach and baselines**. Unlike previous experiments, here we use a supervised text-only model (Su et al., 2022b) finetuned on text-only ROCStories (Mostafazadeh et al., 2016) dataset. Hence, we can safely assume that this specialist text-only model knows the language "better" than the VLM in story generation. We include both visually-conditioned (MAGIC (Su et al., 2022a)) and text-only (Contrastive search (Su and Collier, 2023)) baselines. Refer to appendix B for more details on the baseline results.

**Results**. Table 6 presents the results of open-ended story generation. VLIS outperforms both Contrastive Search and MAGIC in all metrics. While Naïve Ensemble builds more diverse text (rep-2 and div.), its severely low coherence score suggests that its stories are less consistent, as represented in qualitative samples of appendix G. Fi-

(a) **Question**: What kind of dog is in this picture?
**GT**: rottweiler
**VLIS**: rottweiler
**BLIP-2**: a dog
**OPT**: pug
**Naïve Ensemble**: a dog

(b) **Question**: Which one of these animals is native to north america?
**GT**: deer
**VLIS**: deer
**BLIP-2**: zebra
**OPT**: wolf
**Naïve Ensemble**: zebra

(c) **Question**: What material is burning?
**GT**: wax
**VLIS**: paper
**BLIP-2**: umbrella
**OPT**: wood
**Naïve Ensemble**: the material is burning

(d) **Question**: What ocean is being surfed?
**GT**: atlantic, pacific
**VLIS**: water
**BLIP-2**: the ocean
**OPT**: ocean
**Naïve Ensemble**: the ocean

Figure 4: Generation results in the OK-VQA dataset (Marino et al., 2019). We color the intention of the question **green** and answers that defy such intention with **red**. (c) and (d) are failure cases.

nally, while the base VLM (BLIP-2) shows high image-text correspondence as reflected in high CLIPScore, it cannot generate an articulate story as its low performance on other scores shows.

## 5 Qualitative Results

**Commonsense Understanding**. Figure 4 illustrates zero-shot results in the OK-VQA dataset (Marino et al., 2019). In (a) and (b), the baselines including the base VLM and Naïve Ensemble fail to understand the intention of the question (*kind of dog* and *native to north america*). While the text-only model understands the question better and suggests plausible answer candidates (*pug* and *wolf*), it has no access to the visual inputs and ultimately outputs an incorrect answer. On the other hand, VLIS sensibly combines commonsense reasoning and visual context.

Results for images (c) and (d) depict the failure cases. In (c), VLIS follows the reasoning process of the text-only language model to deduce that the answer should be a type of material. However, as the VLM focuses on the frontal object (*umbrella*), VLIS wrongly concludes the answer is the mate-

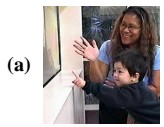

(a) *Hey, It's me*
VLIS:, your son's teacher, and I'm going to show you a picture of me and your son.
**BLIP-2**:, person.

*Today I went*
VLIS: to the library to pick up a book for my son and found out that they have a special section for children with special needs.
**BLIP-2**: to the museum with my son.

*Here is a romantic message. Answer:*
VLIS: I love you and I want to spend the rest of my life in front of you smiling and touching
**BLIP-2**: love you mom

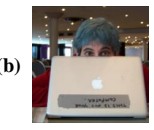

(b) *Hey, It's me*
VLIS: your friend. I have an apple computer and I'm going to be at the convention in a few weeks.
**BLIP-2**: cynthia, I'm a computer scientist.

*Today I went*
VLIS: to an Apple store. I bought an iMac, a keyboard, and a trackpad for my wife's computer.
**BLIP-2**: to a conference.

*Here is a romantic message. Answer:*
VLIS: Dear John, I love you. You are the apple of my eye. Yours truly
**BLIP-2**: I love you.

Figure 5: Open-ended generation results with BLIP-2 (Li et al., 2023b) as the base VLM. We use three text prompts (*Hey, It's me*, *Today I went*, and *Here is a romantic message. Answer:*) to test whether VLIS can actively adjust its response according to the text prompt while maintaining visual alignment.

rial of that object (*paper*, which is coincidentally a flammable material as well). In (d), the text-only model produces an incoherent output (*ocean*). VLIS inherits this misinterpretation and likewise generates an incorrect answer (*water*). In conclusion, VLIS induces coordination of the VLM's visual specificity and the text-only model's commonsense understanding but carries on the modeling insufficiency of the individual modalities.

**Open-Ended Generation**. Lastly, we demonstrate the open-ended generation capability of VLIS in Figure 5. Here, VLIS should condition its output on the diverse text prompt and the image. Unlike the base VLM, it clings tighter to the prompt and produces realistic self-introduction (*hey, it's me*), personal journal (*today I went*), and romantic messages (*here is a romantic message. answer:*). Also, VLIS plays pun on the word *apple* (see *apple laptop* in the image and *apple of my eye*). Refer to appendix G for more baseline samples.

## 6 Related Work

**Combining VLMs with text-only LMs**. Early large-scale VLMs (LXMERT (Tan and Bansal, 2019), VisualBERT (Li et al., 2019) and ViL-BERT (Lu et al., 2019)) saw the benefits of text-only pretraining by initializing their text encoder with a masked language model BERT (Kenton and Toutanova, 2019). Later, Frozen (Tsimpoukelli et al., 2021) started a trend of freezing the language model and learning only the vision-language relationship. More recent models such as Flamingo (Alayrac et al., 2022) and BLIP-2 (Li et al., 2023b) also freeze the image encoder. ESPER (Yu et al., 2022) uses reinforcement learning to combine image encoders with language models.

Better aligned with our approach are decoding-oriented methods for image-conditioned text generation. ZeroCap (Tewel et al., 2022) uses the gradient signal from a pretrained image-text alignment scorer (CLIP (Radford et al., 2021)) to update the language model's memory. Magic (Su et al., 2022a) also utilizes CLIP. Unlike ZeroCap and Magic, VLIS utilizes autoregressive VLMs (Li et al., 2023b), rather than CLIP.

**Language Model Decoding**. Language model decoding is the process of generating text from a pretrained language model. Traditional decoding methods use greedy decoding and beam search to find the most likely sequence of words. The truncated sampling algorithms such as Top K sampling (Fan et al., 2018; Holtzman et al., 2018; Radford et al., 2019), Nucleus sampling (Holtzman et al., 2020), and Typical P sampling (Meister et al., 2022) have been proposed to avoid text degeneracy. Recent deterministic algorithms, such as Contrastive decoding (Li et al., 2022b) and contrastive search (Su et al., 2022b; Su and Collier, 2023), provide a better trade-off between text fluency and model likelihood. Neurologic (Lu et al., 2021) and Neurologic A*esque decoding (Lu et al., 2022b) control the language models to include given words in their outputs. As shown in the experiments, VLIS can be used jointly with any decoding method, including beam search and contrastive search.

## 7 Conclusion

We propose VLIS, a novel framework to alleviate the language modeling burden of visual-language models (VLMs). VLIS combines the linguistic understanding capability of the text-only language models with the visual conditioning strength of the VLMs by importance sampling. To isolate the VLMs' visual conditioning power, VLIS uses pointwise mutual information to suppress their text-only marginal distribution. Our framework

enhances the base VLM in commonsense reasoning (WHOOPS (Bitton-Guetta et al., 2023), OK-VQA (Marino et al., 2019), and ScienceQA (Lu et al., 2022a)) and complicated text generation (Concadia (Kreiss et al., 2022), Image Paragraph Captioning (Krause et al., 2017), and ROCStories (Mostafazadeh et al., 2016)) problems. In the future, VLIS can be extended to incorporate other modalities for which the paired multimodal data is even scarcer. We hope that VLIS sparks an interest in better utilization of off-the-shelf multimodal pretrained models.

## 8 Ethical Considerations & Limitations

**Potential ethical concerns**. As an inference time method, VLIS inherits some known problems of both the VLMs and the unimodal text-only language models as well:

- Hallucination: VLMs are known to hallucinate information absent in the training data (Rohrbach et al., 2018). While VLIS may strengthen visual conditioning and thereby contribute to reducing the rate of visual hallucination, completely eradicating it is beyond the scope of this research.

- Social bias: It is widely known that VLMs reflect or even amplify (Hendricks et al., 2018; Hirota et al., 2022) social bias (*e.g.* gender or race) in the training data. We have yet to determine how VLIS affects social bias in the base models. Thus, outputs generated using VLIS may contain social bias.

It is a meaningful direction to combine VLIS with reinforcement learning (Ramamurthy et al., 2023; Yu et al., 2023) or reward-based decoding algorithm (Su et al., 2022a) to alleviate the problems above, but we leave that to future research.

**Limitation of VLIS and future work**. Firstly, we acknowledge that this paper only explores a small fraction of the possible combinations of text-only models and VLMs. A large-scale wide search in this regard would reveal 1) the better-performing pairs of text-only LM and VLM and 2) the required characteristics of a good model pair.

Secondly, VLIS could be extended to more modalities than the image-to-text generation problem covered here. Other modalities, such as audio and document may also benefit from applying VLIS to their modality-specific foundational model.

Finally, VLIS can be short-sighted. The method combines the outputs of the VLMs and the text-only models at the very last stage of token likelihood. As a result, VLIS score might be misleading when both models assign high probabilities to the same token for different reasons (*e.g.* homophones). It may help to estimate scores for the future generated text by rolling out a few generative steps and aggregating the output (Lu et al., 2022b), which we leave to future works.

## 9 Acknowledgement

This work was supported by Institute of Information & communications Technology Planning & Evaluation (IITP) grant funded by the Korea government (MSIT) (No.2020-0-01361), Institute for Project-Y, and NCSOFT Vision/NLP Center.

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

## A VLM Failure Cases

### A.1 Landmark Recognition Experiment

To better understand the named entity recognition problem in VLMs' image descriptions, we check whether their descriptions for pictures of popular landmarks contain the proper names. We first collect the names of the 100 most popular landmarks [2]. Then, we filter the list by removing names of landmarks without proper nouns (*e.g.* *Middle of the Earth*), keeping 80 landmarks in total. Finally, we download the corresponding pictures from Wikipedia. Given the prompt *What is this?*, we task the VLM to generate a response as long as 100 tokens and check whether the output contains the name of the given landmark. Note that some landmarks have alternative names. Hence, we collect alternative names from Wikipedia and count the model-generated answer as correct when it contains any of the possible names. Finally, we check whether the model tried to answer or not by inspecting whether the model-generated text contains the name of any landmark in our list. We calculate the precision score by dividing the number of correct predictions by the number of tries.

Our landmark dataset [3] is tiny compared to the similar dataset (Weyand et al., 2020) for a purpose: we want to check whether the VLM avoids telling the named entities, not whether the VLM saw them in the training process. Hence, we narrow the scope of evaluation to the most popular landmarks, in which we can assume that most of the entity names are found in the VLM training dataset.

Table 7 and Figure 6 compare base LLAVA (Liu et al., 2023) and VLIS in our landmark recognition dataset. The result shows that the VLM (LLAVA) knows at least about half the landmarks' names, but does not tell them without applying VLIS. Also, VLIS shows good precision, showing that it does not get more correct answers by guessing more. We further demonstrate that a proper answer to our prompt *What is this?* should contain the name of the landmarks: when we present GPT3 with the ground-truth alt captions and the prompt, GPT3 always includes the landmark names in its output.

### A.2 More Qualitative Results

Figure 7 shows full raw text outputs for the VLM failure cases shown in Figure 1. Figure 8 depicts

---

[2] The list for landmarks is from www.listchallenges.com/100-most-famous-landmarks-around-the-world.

[3] We will release the dataset to the public.

| Models | GT Caption | Acc | Prec |
|--------|:----------:|-----|------|
| GPT3 | ✓ | 1.00 | 1.00 |
| LLAVA | | 0.16 | 0.48 |
| Ours | | **0.41** | 0.70 |

Table 7: Results on our landmark recognition experiment. *Acc* denotes accuracy and *Prec* denotes precision.

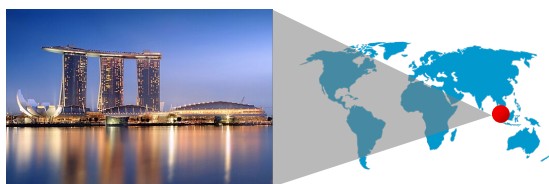

**Ours:**
At nighttime, the **Marina Bay Sands** building, also known as the hotel tower, is floodlit and its surrounding harbor is bustling with activity.
**LLAVA**:
The image features a stunning view of a large building situated on the water. The building appears to be a hotel or a resort, and it is connected to a nearby island by a bridge.

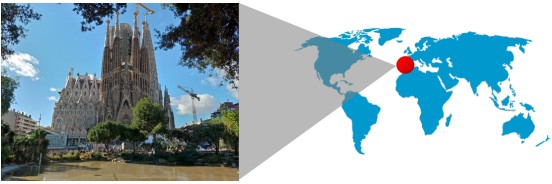

**Ours:**
The structure in the pictures is the well-known **Sagrada Familia basilica** in Barcelona, Spain.
**LLAVA**:
the image features a large cathedral with a tall tower and a steeple, which is likely a famous landmark in the city.

Figure 6: Comparison of LLAVA and VLIS in the landmark recognition experiment.

more samples for the failure case 2: the base VLM (LLAVA) is distracted by misleading visuals while VLIS does not.

## B Implementation Details

**Computational Requirements**. Using LLM.int8 approximation (Dettmers et al., 2022), a single NVIDIA TITAN RTX GPU (24GB Memory) fits both the BLIP-2 2.7B and OPT 1.3B models. Flan-T5 XL and XXL models need more memory and VLIS using the larger backbones requires NVIDIA A6000 GPU (48GB) for inference. Both LLAVA 13B and Vicuna 7B fit into an A6000 GPU at the same time. Generating 50 tokens takes $\sim 20$ seconds in all settings.

**Hyperparameters**. We fix the fluency threshold $\alpha = 0.001$ in all experiments and use beam search with beam size 5. For QA problems, we

apply length penalty $< 0$ on the beam score to induce succinct answers following the literature (Li et al., 2023b). The opposite behavior is required for longer text generation, so we set the value larger than $0$ for open-ended generation problems. The language temperature $\tau$ is manually selected by examining the text quality of three samples per task.

**Task-Specific Hyperparameters**. For VQAv2 (Goyal et al., 2017), OK-VQA (Marino et al., 2019), and ScienceQA (Lu et al., 2022a) datasets, we set the language temperature $\tau = 1.25$ and length penalty $-1.0$ to induce succinct answers generated with stronger visual conditioning. In Concadia (Kreiss et al., 2022), $\tau = 0.67$ and length penalty $-2.0$ is used for succinct caption-style text with better text conditioning. For Image Paragraph Captioning (Krause et al., 2017) experiments we use $\tau = 0.67$ and length penalty $1$ to induce longer captions. Also, we apply contrastive search (Su and Collier, 2023) with a penalty of $0.6$ to avoid text degeneracy.

**Flan-T5 Hyperparameters**. For the backbone comparison study in appendix E, we set the VLM backbone to BLIP-2 Flan-T5 (Li et al., 2023b) and text-only model to Flan-T5 (Chung et al., 2022). For Flan-T5 variants, we compensate the overconfidence of the model with a large temperature of $1.5$ to normalize the logit outputs. For the same reason, we also relax the fluency threshold $\alpha = 0.0001$. Finally, the language temperature $\tau$ is set to $0.9$.

**Baseline Hyperparameters**. We share the same hyperparameters as in VLIS for all our implemented baselines; LLAVA, BLIP-2, OPT-IML, and Naïve Ensemble. We do not modify the beam size $5$ and fluency threshold $\alpha = 0.001$, and change the length penalty accordingly to the task following the VLIS hyperparameters.

**Few-Shot Settings**. For Image Paragraph Captioning (Krause et al., 2017), we use three ground-truth examples to prime the models for the paragraph-long generation task. However, one cannot provide multiple images as inputs to the backbone VLM model (BLIP-2 (Li et al., 2023b)). Hence, we simply insert the few-shot samples in the text domain and provide only the single target image as the visual context.

**uint8 Inference**. LLM.int8 (Dettmers et al., 2022) is an approximated inference technique for large language models. It applies vector-wise quantization and mixed-precision decomposition to reduce memory consumption without performance

| Models | Random | # Images | OKVQA |
|---|---|---|---|
| VLM-only | | | 31.7 |
| Ours | **False** | **2** | **34.2** |
| Ours | True | 1 | 29.0 |
| Ours | True | 2 | 32.2 |
| Ours | True | 10 | 35.3 |

Table 8: Results in the OK-VQA validation set. Our default option (prefined set with two images) is marked bold.

degradation. We employ the technique to jointly run both text-only LM and VLM on a single GPU.

**Randomness**. As VLIS is a deterministic inference time algorithm, no randomness is involved in any of the experiments. A stochastic sampling version of VLIS may require variance analysis, but we leave that to future research.

**Evaluating Story Generation**. While the official repository of MAGIC (Su et al., 2022a) shares the inference results, it does not contain the evaluation scripts. Thus, we consult the repository Contrastive Decoding (Li et al., 2022b) for the evaluation script for an open-ended generation problem. Due to the difference in the evaluation code, our baseline scores are different from the results reported in MAGIC (Su et al., 2022a). However, we still use the public inference results for the baselines and evaluate each model with a publicly available code, making our evaluation pipeline unbiased, transparent, and reproducible.

## C Marginal Approximation Experiment

In the main paper, we propose using one or two images with minimal visual information (black-filled and white-filled) as a functional candidate with minimum computational overhead. To investigate the alternative approaches, we conducted an additional experiment in the OK-VQA dataset. The variables considered here are 1. Random vs. predefined (black-filled and white-filled) set of images and 2. The number of images used to approximate the expectation. We keep everything else the same as in Table 2 and only adjust the marginal approximation scheme.

Our results are summarized in Table 8. First, a random set of images is inferior to our predefined set of images for approximating the marginal. Second, 10 random image set offers a better approximation than the predefined set of two images. Still, the 10 random images option requires 11 passes

| Models | $\alpha$ | OKVQA |
|---|---|---|
| VLM-only | | 31.7 |
| Ours | 1e-1 | 13.8 |
| Ours | 1e-2 | 30.1 |
| **Ours** | **1e-3** | **34.2** |
| Ours | 1e-4 | 34.4 |
| Ours | 1e-5 | 33.1 |
| Ours | 0 | 32.3 |

Table 9: Results in the OK-VQA validation set. Our default fluency threshold value ($\alpha = 1e - 3$) is marked bold.

| VLM Backbone | LLM Backbone | Ours | Vanilla | | Naïve Ensemble |
|---|---|---|---|---|---|
| | | | LLM | VLM | |
| $OPT_{2.7B}$ | $OPT_{1.3B}$ | 34.2 | 19.1 | 31.7 | 26.6 |
| $F\text{-}T5_{XL}$ | $F\text{-}T5_{Base}$ | 29.8 | 12.5 | 40.7 | 34.4 |
| $F\text{-}T5_{XL}$ | $F\text{-}T5_{XL}$ | 43.4 | 19.3 | 40.7 | 39.0 |
| $F\text{-}T5_{XL}$ | $F\text{-}T5_{XXL}$ | 43.9 | 21.3 | 40.7 | 42.0 |
| $F\text{-}T5_{XXL}$ | $F\text{-}T5_{XXL}$ | 47.5 | 21.3 | 45.9 | 44.4 |

Table 10: Backbone comparison experiments on the validation set of the OK-VQA dataset (Marino et al., 2019). F-T5 denotes T5 trained on FLAN dataset (Wei et al., 2022).

of VLM per token generation, making it largely inefficient for practical usage.

## D Fluency Threshold Experiment

Here, we examine the effect of fluency threshold value $\alpha$ on the generation quality of VLIS. This experiment extends the OK-VQA commonsense reasoning experiment in Table 2 and keeps all other variables the same except for $\alpha$.

Table 9 shows that VLIS consistently outperforms the VLM-only baseline for all values of $\alpha$ in the range of $[1e - 3, 1e - 5]$. Too large values ($[1e - 1, 1e - 2]$) still harm the performance as they typically leave only one or two token candidates for the VLIS Score to choose from.

## E Backbone Scale Experiment

We conduct a comparison study to test whether the improvement offered by VLIS is generalizable to a wider set of architectures and model sizes. Here, we mainly evaluate VLIS with Flan-T5 variants as both the text-only LM and VLM backbones. T5 (Raffel et al., 2020) is an encoder-decoder transformer unlike the decoder-only autoregressive language models (*e.g.* OPT (Zhang et al., 2022) and GPT-3 (Brown et al., 2020)). Flan-T5 (Chung et al., 2022) further trains T5 for better responsiveness in instruction prompts. Table 10 summarizes the backbone comparison results on the OK-VQA dataset (Marino et al., 2019). In all combinations of model sizes except for FlanT5$_{Base}$, VLIS improves the commonsense reasoning capability of the VLM backbone. Also, Naïve Ensemble performs unreliably depending on the choice of the text-only LM and performs worse than the VLM itself in most of the settings. The FlanT5$_{Base}$ LM makes VLIS work worse than the VLM. Since VLIS is built on the assumption that the text-only LM knows the

human language distribution better than the VLM, this deterioration of performance further supports our explanation of why VLIS works.

## F Prompt Templates

In the prompt templates below, TLM denotes the prompt presented to the text-only model and VLM denotes that given to the VLM.

- **OK-VQA & VQAv2**
    - Variables:
      `[QUESTION]`
    - TLM
      `Question: [QUESTION] Answer:`
    - VLM
      `Question: [QUESTION] Answer:`

- **ScienceQA**
    - Variables:
      `[QUESTION], [CONTEXT], [CHOICES]`
    - TLM
      `Answer the multi-choice question given the image. Question: [QUESTION] [CONTEXT] Choices: [CHOICES] Answer:`
    - VLM
      `Answer the multi-choice question given the image. Question: [QUESTION] [CONTEXT] Choices: [CHOICES] Answer:`

- **Concadia**
    - TLM
      `Write a short caption that describes the image. Article: "[ARTICLE]" Caption:`
    - VLM
      `The image describes`

- **Image Paragraph Captioning**
  - Variables:

    `[ARTICLE]`
  - TLM

    ```
    Write a multi-sentence long
    paragraph describing the image.
    Each sentence should describe a
    different aspect of the image and
    should be concise.\n
    (Image 1) Image Description:
    [Description Sample 1]\n
    (Image 2) Image Description:
    [Description Sample 2]\n
    (Image 3) Image Description:
    [Description Sample 3]\n
    (Image 4) Image Description:
    ```
  - VLM

    ```
    Write a multi-sentence long
    paragraph describing the image.
    Each sentence should describe a
    different aspect of the image and
    should be concise.\n
    (Image 1) Image Description:
    [Description Sample 1]\n
    (Image 2) Image Description:
    [Description Sample 2]\n
    (Image 3) Image Description:
    [Description Sample 3]\n
    (Image 4) Image Description:
    ```

- **ROCStories**
  - Variables:

    `[TOPIC]`
  - TLM

    ```
    [TOPIC] <|endoftext|>
    ```
  - VLM

    ```
    The image describes
    ```

- **WHOOPS**
  - TLM

    ```
    Decide whether the image is
    strange or natural in terms of
    physics, commonsense, or etc.\n
    Start with "The image shows"
    ```
  - VLM

    ```
    Decide whether the image is
    strange or natural in terms of
    physics, commonsense, or etc.\n
    Start with "The image shows"
    ```

## G  More Qualitative Samples

We include more qualitative samples for image paragraph generation in Figure 9, story generation in Figure 10, and open-ended generation in Figure 11.

## (a) Describing named entities

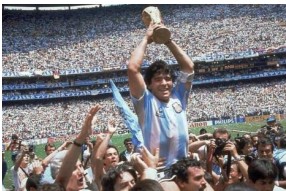

**Who is this?**

**BLIP-2**
person, **argentina's greatest player of all time**
**LLAVA**
The image features a soccer player who has just won a championship, as he is lifted up by a crowd of people celebrating the victory. The soccer player is holding a trophy, signifying the achievement.
**Ours**
Sources suggest that the person being described is **Diego Maradona**, a professional footballer who won the **Golden Boot award at the 1982 World Cup**. In the image, Maradona is being lifted above a stadium crowd

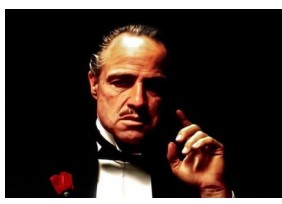

**Does he care for his family?**

**BLIP-2**
**yes**
**LLAVA**
While I cannot determine the man's intentions or emotions as an AI, the image shows a man in a suit, who appears to be smiling and has his fingers crossed. This body language might suggest that he is confident
**Ours**
**Michael Corleone**, also known as Vito Andolini Corleone, the character depicted in the **Godfather film series**, is shown to **care deeply for his immediate family**.

## (b) Visual distractors for simple facts

**[Image of a Seagull]**

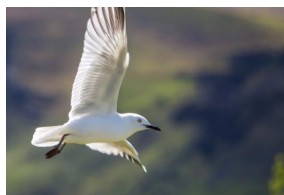

**Can ostriches fly?**

**GPT3**
**No**, ostriches are not capable of flying. They are the largest living species of bird and are known for their long, powerful legs and neck, which they use to run at high speeds, but they do not have the physical characteristics
**BLIP-2**
**yes**, but they are not very good at it
**LLAVA**
**Yes**, ostriches can fly. Although they are large birds, they are capable of taking flight using their powerful wings. Ostriches have strong muscles and a high lift-to-drag ratio, which enables them to become air
**Ours**
**No**, ostriches are not capable of flying. They are ground-dwelling birds that are known for their long, powerful legs and fast running speed. In the wild, ostriches migrate by running long distances, but

**[Image of a Monkey]**

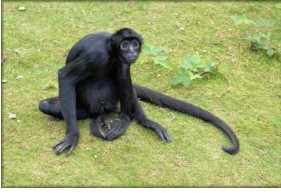

**Do chimpanzees have tails?**

**GPT3**
**No**, chimpanzees do not have tails. They are one of the few primates that lack a tail. Chimpanzees belong to the family Hominidae, along with humans, bonobos, gorillas, and orangutans.
**BLIP-2**
**yes**.
**LLAVA**
**Yes**, chimpanzees do have tails. Their tails are short and usually held upright or curled up over their backs. In the image, the chimpanzee is walking on a tree branch with its tail in
**Ours**
Chimpanzees **do not have tails**. Chimpanzees are primates and are closely related to humans. They are also known as great apes and are characterized by their large brains, hands and feet, and the ability to

Figure 7: Raw text output for samples described in Figure 1.

## Visual distractors for simple facts

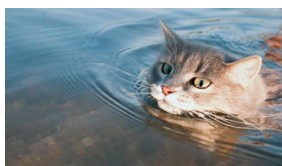

**Do cats swim?**

**GPT3**

While cats are generally known for their aversion to water, some cats are capable of swimming. However, **swimming is not a natural behavior for most cats**, and they usually prefer to avoid water whenever possible.

**LLAVA**

**Yes, cats can swim.** In the image, a gray cat is swimming in a body of water, demonstrating that cats have the ability to navigate and move efficiently through water.

**Ours**

Some cats may be able to swim to some extent, but **they are not natural swimmers** and are not as comfortable in water as dogs are. In the wild, cats typically avoid water and do not have the same instinctive swimming capability

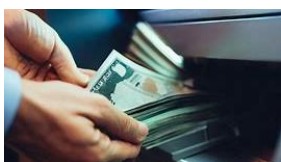

**Can I withdraw cash from a blood bank?**

**GPT3**

**No**, you cannot withdraw cash from a blood bank. Blood banks are facilities that collect, process, and store donated blood for medical use.

**LLAVA**

**While it is possible to withdraw cash from a blood bank,** the context of the image suggests that the person is actually taking money out of an ATM (Automated Teller Machine).

**Ours**

**No**, it is not possible to withdraw cash from a blood bank. Blood banks are typically medical facilities that collect and process blood donations, and they do not provide cash withdrawals.

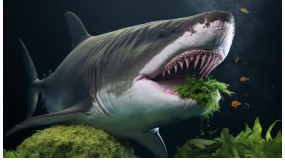

**Are sharks herbivores?**

**GPT3**

**No**, sharks are not herbivores. Sharks are carnivorous predators, meaning they primarily feed on other animals.

**LLAVA**

**Yes**, sharks are generally considered herbivores. They primarily feed on a variety of plants, including algae, as well as small animals and fish. In the image, the shark is consuming sea vegetation, which is a

**Ours**

**No**, sharks are not herbivores. In the wild, sharks primarily feed on fish, squid, and other marine animals. However, in aquariums and marine parks, it is common for sharks to be fed

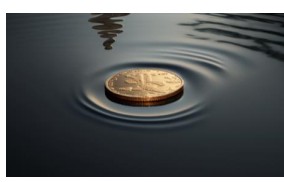

**Do coins float on water?**

**GPT3**

In general, most coins made of metals such as copper, nickel, or zinc alloys will **sink in water** due to their higher density. However, if a coin has a very small size or a unique shape that allows it to trap air, it may float momentarily or for a short period of time before eventually sinking.

**LLAVA**

**Yes, coins do float on water.** In the image, a single penny is floating on the surface of a body of water, such as a lake or a pool. This is because the coin is made of materials that are denser than water.

**Ours**

**Coins do not float on water.** In fact, they will usually sink to the bottom. This is because coins are made of a material that is denser than water, so they will tend to sink rather than float. In the case of

Figure 8: More examples in which the visual context deceives the base VLM (LLAVA (Liu et al., 2023)) into denying simple facts.

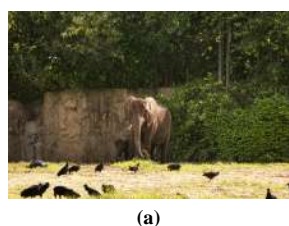

**(a)**

**GT**: Two elephants are standing together outside by a concrete wall. One elephant is considerably smaller then the other one. Right by the elephants there is a large amount of bushes and trees with green leafs. The ground is mostly dirt with patches of green grass throughout. On the grass eating there is about a dozen or so large black crows.
**Ours**: An elephant in an enclosure surrounded by crows and pigeons is looking around for something to eat. The elephant has a trunk in its mouth and is sniffing the ground. The ground around the elephant is covered in dirt and grass. The elephant is standing on its tusks.
**BLIP-2**: An elephant stands in the middle of a field of grass. The grass is green and lush. The elephant has a brown trunk and a brown face. The elephant has a brown trunk and brown face. The elephant has a brown trunk and brown face.
**OPT**: A young man is sitting on a park bench. He is wearing a black t-shirt and blue jeans. He is holding a baseball bat in his right hand. He is throwing the ball from a pitcher's mound. The pitcher has a uniform on that is grey and blue. His cap is black with a fork shaped symbol.
**Naïve Ensemble**: A young man is sitting on a bench in a park. He is wearing a blue shirt and blue jeans. He is holding a baseball bat in his right hand. He is looking at the ball that is being thrown from a pitcher's mound. The pitcher has a uniform on that is grey and blue. His cap is black with a fork shaped symbol.

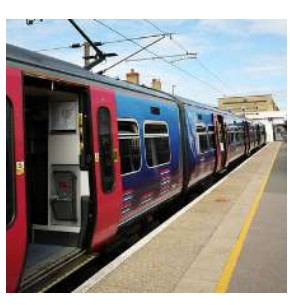

**(b)**

**GT**: A train is opening it's doors. The train is currently parked at a train station. The train is blue, the doors are red, and it has white stripes on it. There is a long yellow line near the train area.
**Ours**: A passenger train is parked at a train station. The train is red in color with blue and white stripes on it. There are power lines overhead that are connected to the train. The train is pulling into the station. A man is sitting on the train with his back to the camera.
**BLIP-2**: The man on the skateboard is leaning his body or balance. He is in motion riding along the road. He is wearing a grey helmet with a print on it. There is an orange cone in the street that he will have to go around.
**OPT**: A young man is sitting on a park bench. He is wearing a black t-shirt and blue jeans. He is holding a baseball bat in his right hand. He is throwing the ball from a pitcher's mound. The pitcher has a uniform on that is grey and blue. His cap is black with a fork shaped symbol.
**Naïve Ensemble**: The man on the skateboard is leaning his body or balance. He is in motion riding along the road. He is wearing a grey helmet with a print on it. There is an orange cone in the street that he will have to go around.

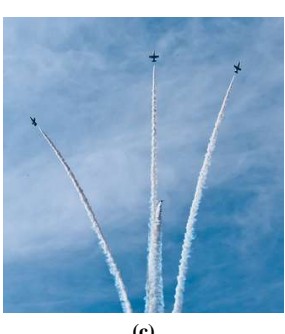

**(c)**

**GT**: Four airplanes are flying in the sky in order to put on a show. The planes are dark colored and rather small. Each plane is flying straight up into the sky and a large plume of white exhaust is leaving a trail from the planes. The trail of exhaust shows that the planes started our flying up close together, but they are now separating and flying away from each other. The sky in the background is blue with a layer of light white clouds.
**Ours**: Four planes fly through the blue sky with smoke trailing behind them. They are performing for the public. The planes are flying in formation. One of the planes is in the middle of the formation and the other three planes are in front of the plane in the middle of the formation.
**BLIP-2**: An airplane is flying in the sky. The airplane has a red, white, and blue color scheme. The airplane is flying in a blue sky. The airplane is flying in a clear blue sky. The airplane is flying in a clear blue sky.
**OPT**: A young man is sitting on a park bench. He is wearing a black t-shirt and blue jeans. He is holding a baseball bat in his right hand. He is throwing the ball from a pitcher's mound. The pitcher has a uniform on that is grey and blue. His cap is black with a fork shaped symbol.
**Naïve Ensemble**: A young man is sitting on a bench in a park. He is wearing a blue shirt and blue jeans. He is holding a baseball bat in his right hand. He is wearing a baseball cap on his head. He is looking up at the sky with his left eye. He is looking down at the ground with his right eye.

Figure 9: Generation results on the Image Paragraph Captioning dataset (Krause et al., 2017). The correct visual expressions are marked in **green** and wrong ones are in **red**.

**Title:** Joanne Went Sky Diving

**GT**: Joanne always wanted to go skydiving but was hesitant. Joanne was hesitant to go because she was afraid of heights. Joanne decided to face her fears head on and go skydiving. On her way in the car, she listened to music and took deep breaths. Joanne went sky diving and was very proud of herself.

**Ours**: Joanne went skydiving with her friends. She had a great time on the water at the end of the day. When it was time to take off the parachute was broken. Joanne's friends had to call for help to get it fixed.

**Naïve Ensemble**: Joanne was at the beach with her friends. She decided to go on a dive. When she got to the water, she saw her friends in the water. Joanne jumped out of the water and ran to the other side of the water. She had no idea what was going on until she was rescued by rescue workers.

**MAGIC**: Joanne was at the beach with her friends. She decided to jump off the sand to see what was out there. When she landed, she saw a huge wave. The wave was so big that she had to stop and get help. Joanne was able to land safely and return to the beach.

**Contrastive Search**: Joanne was at the beach with her friends. She decided to go on a dive. When she got to the water, she saw her friends in the water. Joanne jumped out of the water and ran to the other side of the water. She had no idea what was going on until she was rescued by rescue workers.

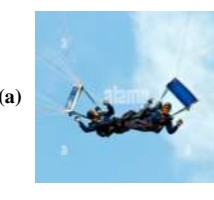

**(a)**

**Title:** Jobs

**GT**: Lana liked to spend money, but she didn't have a job. One day, her dad made her find a job so she could spend her own money. Lana found a job, but didn't work very hard. Eventually, Lana got fired. Her dad stopped giving her money to spend.

**Ours**: Steve wanted a job. But he didn't know what to do with his money. So he went to the bank and asked for one. And it turned out to be very cheap. Steve started his own company that same year.

**Naïve Ensemble**: Steve wanted to start a career in his spare time after high school. So he went to college and got his bachelor's degree at the age of 18. But he didn't know what he wanted to do with his free time. He decided to become an electrician for the city of New York.

**MAGIC**: The man went to work. He was asked to join a new job. He did not like the job. He quit the job. His boss found out that he had quit.

**Contrastive Search**: The man went to work. He wanted to be a manager. He asked his boss for help. His boss said he had no idea. The man quit his job.

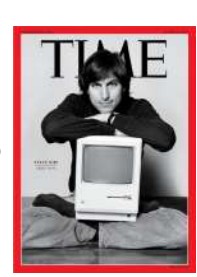

**(b)**

**Title:** Looking for peace

**GT**: Nina was dreaming of world peace her whole live. She knew she couldn't change the world but made small changes. She contributed to her community and then to her country. She came up with so many good ideas, she was congratulated. She was proud when she received a Nobel prize for her work.

**Ours**: Yesterday I went for a walk in the woods. It was dark and windy so I couldn't see anything out of the corner of my eye. After walking through the woods for an hour I found peace. I decided to return home and look for peace in the future.

**Naïve Ensemble**: A girl with long hair walked into my yard. She wanted to talk to me but I didn't see her in front of me. I asked her if she was looking for someone and she said yes. The next day my neighbor called me to tell me her hair was in bad shape

**MAGIC**: The man walked into a house. He saw an empty lot. He asked his neighbors to look for peace. The neighbors were skeptical. It was not enough to solve the problem.

**Contrastive Search**: The man went into a house. He saw an old woman sitting on the porch. She was crying. The man asked her to leave. Her mother took her to the police station.

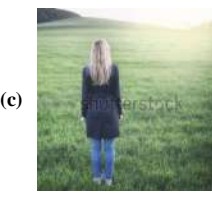

**(c)**

Figure 10: Generation results on the ROCStories dataset (Mostafazadeh et al., 2016). Expressions showing alignment to the title are colored in **green**, alignment to the image in **blue**, and misinterpretations in **red**.

**(a)**

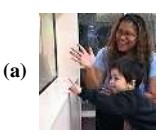

*Hey, It's me*

**Ours**:, your son's teacher, and I'm going to show you a picture of me and your son.
**BLIP-2**:, person.
**OPT**: your brother. Hey it's me, your sister.
**Naïve Ensemble**: , the guy who wrote the article about the guy who wrote the article about the guy who wrote the

*Today I went*

**Ours**: to the library to pick up a book for my son and found out that they have a special section for children with special needs.
**BLIP-2**: to the museum with my son.
**OPT**: to the gym for the first time in two weeks. It felt good to be back in the swing of things.\nThat's awesome! What did you do?
**Naïve Ensemble**: to the doctor's office for the first time in a long time, and the doctor's office is in the same building as the doctor's…

*Here is a romantic message. Answer:*

**Ours**: I love you and I want to spend the rest of my life in front of you smiling and touching
**BLIP-2**: love you mom
**OPT**: I love you and I want to spend the rest of my life with you. You are the best thing that ever happened to me
**Naïve Ensemble**: I love you and I want to spend the rest of my life with you and I want you to

**(b)**

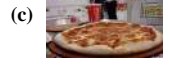

*Hey, It's me*

**Ours**: your friend. I have an apple computer and I'm going to be at the convention in a few weeks.
**BLIP-2**: cynthia, I'm a computer scientist.
**OPT**: your brother. Hey it's me, your sister.
**Naïve Ensemble**: , your friend. I'm here to tell you that you're a douchebag and you should feel bad for being such an asshole to your friends and family.

*Today I went*

**Ours**: to an Apple store. I bought an iMac, a keyboard, and a trackpad for my wife's computer.
**BLIP-2**: to a conference.
**OPT**: to the gym for the first time in two weeks. It felt good to be back in the swing of things.\nThat's awesome! What did you do?
**Naïve Ensemble**: to the library to check out a book, but the librarians wouldn't let me check it out because it was too late in the day. So I had to wait until the next day

*Here is a romantic message. Answer:*

**Ours**: Dear John, I love you.  You are the apple of my eye. Yours truly
**BLIP-2**: I love you.
**OPT**: I love you and I want to spend the rest of my life with you. You are the best thing that ever happened to me
**Naïve Ensemble**: love you and miss you and hope you are having a great time with your family and friends. I love you and miss you and hope you

**(c)**

*Hey, It's me*

**Ours**: and my friend. We were wondering if you would sell us a pizza and a coke at the same time. Would you be able to do that
**BLIP-2**: again!
**OPT**: your brother. Hey it's me, your sister.
**Naïve Ensemble**: again. I just wanted to let you all know that I will be out of town for the next couple of days. I will be out of town

*Today I went*

**Ours**: to a pizza place and got a large pepperoni pizza. It was pretty good. I also got a diet coke
**BLIP-2**: to a pizza place.
**OPT**: to the gym for the first time in two weeks. It felt good to be back in the swing of things.\nThat's awesome! What did you do?
**Naïve Ensemble**: to the grocery store to get some food for dinner. When I got to the deli counter I noticed that there was a very large amount of food

*Here is a romantic message. Answer:*

**Ours**: (Romantic) I would like to take my girlfriend to eat pizza. What do you think is the best pizzeria in your area
**BLIP-2**: pizza on a table.
**OPT**: I love you and I want to spend the rest of my life with you. You are the best thing that ever happened to me
**Naïve Ensemble**: how many pints of cocteau are in the fridge at the end of the work day on friday the thirteenth and saturday

Figure 11: Open-ended generation results with various text prompt. Here we include more baselines than in Figure 5.