# OpenReview forum: "VLIS: Unimodal Language Models Guide Multimodal Language Generation"
_EMNLP/2023/Conference — EMNLP 2023 Main_

### Official Review · Reviewer_z5TT · 2023-07-31

**Soundness:** 4

**Excitement:**

4: Strong: This paper deepens the understanding of some phenomenon or lowers the barriers to an existing research direction.

**Paper Topic And Main Contributions:**

In this research article, the authors introduce a novel approach, referred to as VLIS, aimed at mitigating two prevalent issues associated with the comprehension capabilities of current visual language models (VLMs). The first problem arises when VLMs fail to identify entities in certain scenarios, while the second problem stems from the over-reliance of VLMs on image context, which subsequently results in numerous unsuccessful cases.


The paper's contribution is two-fold. Firstly, a training-free methodology is proposed to extract importance sampling weights from the VLM, providing visual alignment to adjust the token likelihood emanating from the large language model (LLM). Secondly, the proposed methodology exhibits promising performance in various distinct types of tasks.

**Questions For The Authors:**

As stated in the paper, a better visual alignment can be achieved, which may help to reduce the hallucination. Could you provide the results on the image captioning task, which is a less complex text generation task, and provide an analysis related to hallucinations, including the CHAIR scores [1]?

[1] Rohrbach, Anna, et al. "Object hallucination in image captioning." arXiv preprint arXiv:1809.02156 (2018).

**Reasons To Accept:**

1) The method proposed in the study is innovative and intriguing, as it rationally leverages the properties and strengths of both the visual language model (VLM) and the large language model (LLM). Furthermore, its training-free nature significantly cuts down on associated training costs.

2) The performance of the proposed methodology across numerous datasets is promising, which is impressive.

**Reasons To Reject:**

1) As delineated in Appendix E, the inference time is seen to increase due to the requirement of multiple forward passes, which potentially affects the method's efficiency.

2) Despite exhibiting impressive results across several datasets, the proposed method fails to achieve the best outcomes in some, including the ROCStories dataset.

**Reproducibility:**

3: Could reproduce the results with some difficulty. The settings of parameters are underspecified or subjectively determined; the training/evaluation data are not widely available.

**Reviewer Confidence:**

4: Quite sure. I tried to check the important points carefully. It's unlikely, though conceivable, that I missed something that should affect my ratings.

---

> ### Author Rebuttal · Authors · 2023-08-27
>
> We thank the reviewer for acknowledging our method as *innovative and intriguing* and our experiment results *promising and impressive*. We commit to updating the paper draft to better describe the points raised by the reviewer, including the additional experiment results.
>
> ## 1. Regarding reproducibility
>
> We regret that we accidentally omitted the code in the supplementary files in the recent revision. However, we hope that the reviewer reconsiders the reproducibility score on the following grounds.
>
> - Our implementation code was included in the supplementary files of our earlier revisions on the OpenReview website, specifically in the first submission version.
> - We add the main pseudo-code at the bottom of this response.
> - We promise to release the code publicly as soon as the anonymity period ends. Were it not for the anonymity constraint, we would have shared the code link here.
> - VLIS is really simple to implement. It does not require any training or complex pipelining: rather, one only needs the logit outputs for both the text-only LM and VLM.
> - All datasets we used for the experiments are publicly available.
> - All hyperparameters we used are reported in the main paper and the appendix (also the first submission version). Additionally, VLIS introduces only two hyperparameters: $\tau$ and $\alpha$.
> - The GPU requirement is manageable: even VLIS with LLAVA 13B runs on a single NVIDIA A100 (40GB) with num_beams=5. VLIS with BLIP-2 runs on a single 24GB GPU.
>
> ## 2. Running time efficiency
>
> Yes, VLIS introduces additional inference time costs. The default setup of VLIS requires three passes of the VLM with a single pass of the text-only LM.
>
> However, VLIS is considerably more efficient than previous multimodal inference time methods, which require either expensive backpropagation [1] or numerous (~300) passes of the visual model [2] per token generation.
>
> ## 3. Results on the ROCStories dataset
>
> In Table 6 of the main paper, VLIS secures the best scores in Coherence and MAUVE, while achieving the second-best scores in other metrics. We believe this is enough to show that VLIS is better than all other baselines. The reasons are stated below.
>
> Open-ended generation tasks such as ROCStories generation are hard to evaluate; thus we include an extensive set of metrics following previous work [2]. A method with only a high CLIPScore (BLIP-2) just generates image captions (and no stories) as shown in its bad score in all other metrics. Also, while Naive Ensemble performs well in Repetition and Diversity scores, it shows the worst Coherence meaning that it fails to generate fluent text and instead achieves diversity by generating random visually relevant words.
>
> In conclusion, VLIS outperforms the baselines in ROCStories. We saw that the qualitative samples match this conclusion as well; we will include them in the final draft.
>
> ## 4. Image captioning hallucination experiment
>
> We appreciate the reviewer’s insightful suggestion on measuring the image-text alignment strength of VLIS. We thus conducted an additional experiment in the COCO-Captions image captioning benchmark. In the Table below, we show that VLIS reduces object hallucination of both BLIP-2 and LLAVA backbone. Note that this improvement does not come from just mentioning fewer objects overall, VLIS’s better $CHAIR_I$ shows that it sensibly discriminates objects within the image from those that are not.
>
> We’d like to note two auxiliary details here. First, conventional image captioning is not the strongest suit of VLIS since the task doesn’t necessitate superior language modeling capability beyond what’s already present in the VLM itself. Nevertheless, the Mutual Information score of VLIS helps overcome the VLM language bias. Second, our baseline results outperform those reported in a recent paper [3]. Unlike their approach, we confined the generation to 20 tokens and took only the first sentence for a fair comparison between the backbone that generates short captions (BLIP-2) and that with long text (LLAVA) by default.
>
> Table: Image captioning hallucination experiment on COCO validation set (Karpathy split). Lower is better.
> | model  | Decoding |  CHAIR_I  |  CHAIR_S  | Length |
> |---|---|---|---|---|
> | BLIP-2 | Standard | 3.7       | 4.3       | 7.1    |
> | BLIP-2 | Ours     | 2.4       | 3.1       | 7.1    |
> | | | | | |
> | LLAVA  | Standard | 5.4       | 8.0       | 12.6   |
> | LLAVA  | Ours     | 5.0       | 7.2       | 13.3   |
>
> ## 5. References
>
> - [1] ZeroCap: Zero-Shot Image-to-Text Generation for Visual-Semantic Arithmetic, https://arxiv.org/abs/2111.14447
> - [2] Language Models Can See: Plugging Visual Controls in Text Generation, https://arxiv.org/abs/2205.02655
> - [3] Evaluating Object Hallucination in Large Vision-Language Models, https://arxiv.org/abs/2305.10355
>
> ## 6. Pseudo-code
>
> Here is the pseudo-code for the token score calculation part of VLIS. Except for this part, the implementation of VLIS follows the standard language generation code which is widely available via public libraries such as Huggingface.
>
> ```
> <inputs>
> list_marginal_vlm_lprobs  # List of VLM log token probability for each of the marginal images and the previously generated text.
> vlm_lprobs  # VLM log token probability for the input image and the previously generated text.
> lm_lprobs # text-only log token probability for the previously generated text.
> fluency_threshold  # scalar hyperparameter \alpha for fluency masking
> fluency_alpha  # scalar hyperparameter \tau for language model temperature control
> </inputs>
> <code>
> fluency_mask = math.log(fluency_threshold) <= lm_lprobs
> marginal_vlm_lprobs = mean(list_marginal_vlm_lprobs)
> log_visual_score = vlm_lprobs - marginal_vlm_lprobs
> log_vlis_score = lm_lprobs * fluency_alpha + log_visual_score
> log_vlis_score[!fluency_mask] = -math.inf
> </code>
> ```

---

### Official Review · Reviewer_Vw57 · 2023-08-05

**Soundness:** 3

**Excitement:**

4: Strong: This paper deepens the understanding of some phenomenon or lowers the barriers to an existing research direction.

**Paper Topic And Main Contributions:**

This paper introduces Visual-Language models as Importance Sampling weights (VLIS), a novel framework designed to combine the visual conditioning capability of vision-language models (VLMs) with the language understanding of unimodal text-only language models without further training. The main contributions of the paper are:
1. Proposing VLIS, a method that enhances the linguistic understanding of VLMs by using the token likelihoods of unimodal text-only language models and importance sampling weights derived from a VLM for visual alignment.
2. Employing exponentiated pointwise mutual information (PMI) to isolate the visual conditioning capability of the VLMs from their language modeling preferences.
3. Demonstrating the effectiveness of VLIS in various multimodal tasks, including commonsense understanding, complex text generation, and showing that VLIS improves vision-language models on these tasks.
4. Comparing VLIS against several baselines on different tasks and showing its promise as a new direction for multimodal language generation.

Overall, the paper addresses the challenge of enhancing the unreliable linguistic understanding of VLMs and suggests VLIS as a promising approach to combine the strengths of both unimodal and multimodal language models.

**Questions For The Authors:**

Question A: Can you provide more insights into the choice of the fluency threshold value (α=0.001) used in the fluency masking step? How sensitive is VLIS performance to different fluency threshold values?

Question B: In the experiments, you use either one or two images with almost no visual information (black-filled and white-filled) to estimate the VLM marginal likelihood. Have you explored other alternatives or investigated the impact of different image sets on the VLIS performance?

**Reasons To Accept:**

1. The paper proposes VLIS, an innovative method that effectively combines the visual conditioning capability of vision-language models with the language understanding of unimodal text-only language models without additional training. This offers a new direction for multimodal language generation.
2. The paper conducts thorough experiments on various tasks that require both reliable language modeling and strong visual conditioning, such as commonsense understanding, scientific reasoning, and text generation. These experiments demonstrate the effectiveness and versatility of the VLIS framework.
3. The paper compares VLIS against several baselines, including text-only models, VLMs, and naive ensemble approaches, providing valuable insights into the performance of VLIS in different settings and its ability to adaptively fuse commonsense understanding with visual conditioning.
4. The VLIS framework is not limited to specific VLM architectures and can be potentially applied to other multimodal tasks or extended to other modalities, such as audio and documents.
5. The paper provides qualitative and quantitative analyses of the generated outputs, highlighting the improved linguistic capabilities, responsiveness to prompts, and visual conditioning achieved by VLIS compared to the base VLMs.

**Reasons To Reject:**

1. The paper only explores a small fraction of possible combinations of text-only models and VLMs.
2. VLIS combines the outputs of VLMs and text-only models at the very last stage of token likelihood, which might be misleading when both models assign high probabilities to the same token for different reasons.
3. While the paper suggests that VLIS could be extended to other modalities, such as audio and documents, it does not provide experimental results to support this claim.

**Reproducibility:**

3: Could reproduce the results with some difficulty. The settings of parameters are underspecified or subjectively determined; the training/evaluation data are not widely available.

**Reviewer Confidence:**

4: Quite sure. I tried to check the important points carefully. It's unlikely, though conceivable, that I missed something that should affect my ratings.

---

> ### Author Rebuttal · Authors · 2023-08-27
>
> We thank the reviewer for acknowledging that our method *offers a new direction for multimodal language generation* and finding our experiments to be *thorough*.
> We commit to updating the paper draft to better describe the points raised by the reviewer, including the additional experiment results.
>
> ## 1. Regarding reproducibility
>
> We regret that we accidentally omitted the code in the supplementary files in the recent revision. However, we hope that the reviewer reconsiders the reproducibility score on the following grounds.
>
> - Our implementation code was included in the supplementary files of our earlier revisions on the OpenReview website, specifically in the first submission version.
> - We add the main pseudo-code at the bottom of this response.
> - We promise to release the code publicly as soon as the anonymity period ends. Were it not for the anonymity constraint, we would have shared the code link here.
> - VLIS is really simple to implement. It does not require any training or complex pipelining: rather, one only needs the logit outputs for both the text-only LM and VLM.
> - All datasets we used for the experiments are publicly available.
> - All hyperparameters we used are reported in the main paper and the appendix (also the first submission version). Additionally, VLIS introduces only two hyperparameters: $\tau$ and $\alpha$.
> - The GPU requirement is manageable: even VLIS with LLAVA 13B runs on a single NVIDIA A100 (40GB) with num_beams=5. VLIS with BLIP-2 runs on a single 24GB GPU.
>
> ## 2. Regarding reasons to reject
>
> We acknowledge the points raised by the reviewer. However, the main contribution of VLIS is that multimodal inference can benefit from harmonizing existing language models with VLMs. While more backbone combinations, other modalities, and future estimations might strengthen the robustness and generalizability of VLIS, VLIS still works well as-is in the visual-language domain as we show in various experiments.
>
> In conclusion, we hope that the reviewer considers these points as future research directions that VLIS provides instead of grounds to reject the method altogether, as we described in the future works section in our paper.
>
> ## 3. Choice of the fluency threshold
>
> VLIS’s performance is not overly sensitive to the specific value of the fluency threshold, provided it is within a reasonable range. The primary function of Fluency masking is to filter out token candidates that have abnormally high visual weights, which can be attributed to the modeling insufficiency of the VLM.
>
> To show this, we conducted an additional experiment on OKVQA (with BLIP-2) for varying values of the threshold while keeping everything else the same as the main paper. In the Table below, for all values in the range of  ($\[1e-3, 1e-5\]$), VLIS consistently outperforms the VLM-only baseline. Too large values of $\tau$ still harm the performance ($\[1e-1, 1e-2\]$) as they typically leave only one or two token candidates for the VLIS Score to choose from.
>
> Table: Fluency threshold experiment on OKVQA
> | model | $\tau$ | VQAScore |
> |---|---|---|
> |VLM-only| |31.7|
> | | | |
> |Ours|1e-1|13.8|
> |Ours|1e-2|30.1|
> | | | |
> |Ours|1e-3|34.2|
> | | | |
> |Ours|1e-4|34.4|
> |Ours|1e-5|33.1|
> |Ours|0|32.3|
>
> ## 4. Different image sets for the marginal approximation
>
> We offer using one or two images with minimal visual information (black-filled and white-filled) as a functional candidate with minimum computational overhead. To investigate alternative approaches, we conducted an additional experiment on OKVQA. The variables considered here are 1. Random vs. predefined (black-filled and white-filled) set of images and 2. The number of images used for expectation approximation. We keep everything else the same as in the main paper and only vary the marginal approximation scheme.
>
> The Table below shows our results. First, using a single or two random images within the validation set suffers from the expected performance drop. Second, using 10 random images offers a better approximation than the standard scheme (Random Images = False). Still, the 10 random images option requires 11 passes of VLM per token generation, making it largely inefficient for practical usage.
>
> Table: Marginal image set experiment on OKVQA
> | model | Random Images | Num Images | VQAScore |
> | --- | --- | --- | --- |
> |VLM-only|||31.7|
> | | | | |
> |Ours|False|2|34.2|
> | | | | |
> |Ours|True|1|29.0|
> |Ours|True|2|32.2|
> |Ours|True|10|35.3|
>
> ## 5. Pseudo-code
>
> Here is the pseudo-code for the token score calculation part of VLIS. Except for this part, the implementation of VLIS follows the standard language generation code which is widely available via public libraries such as Huggingface.
>
> ```
> <inputs>
> list_marginal_vlm_lprobs  # List of VLM log token probability for each of the marginal images and the previously generated text.
> vlm_lprobs  # VLM log token probability for the input image and the previously generated text.
> lm_lprobs # text-only log token probability for the previously generated text.
> fluency_threshold  # scalar hyperparameter \alpha for fluency masking
> fluency_alpha  # scalar hyperparameter \tau for language model temperature control
> </inputs>
> <code>
> fluency_mask = math.log(fluency_threshold) <= lm_lprobs
> marginal_vlm_lprobs = mean(list_marginal_vlm_lprobs)
> log_visual_score = vlm_lprobs - marginal_vlm_lprobs
> log_vlis_score = lm_lprobs * fluency_alpha + log_visual_score
> log_vlis_score[!fluency_mask] = -math.inf
> </code>
> ```

---

### Official Review · Reviewer_TDrJ · 2023-08-05

**Typos Grammar Style And Presentation Improvements:** n/a
**Soundness:** 3

**Excitement:**

2: Mediocre: This paper makes marginal contributions (vs non-contemporaneous work), so I would rather not see it in the conference.

**Missing References:**

[1] What Matters in Training a GPT4-Style Language Model with Multimodal Inputs?
https://arxiv.org/abs/2307.02469

**Paper Topic And Main Contributions:**

This paper proposes to combine the visual conditioning capability of vision-language models with the language understanding of unimodal text-only language models without further training. Concretely, VLIS extracts pointwise mutual information of each image and text from a visual-language model and uses the value as an importance sampling weight to adjust the token likelihood from a text-only model. VLIS improves vision-language models on several tasks.


**Questions For The Authors:**

n/a

**Reasons To Accept:**

VLIS improves multi-modal LLMs' performances (namely, LLAVA and BLIP-2 OPT 2.7B) on several tasks.

**Reasons To Reject:**

1) The experiments can be improved. The chosen baselines such as LLAVA and BLIP-2 OPT 2.7B are not SoTA multimodal LLMs according to recent study[1]. Concretely, LLAVA generates long texts ignoring the image inputs, while BLIP-2 tends to generate short answers, usually a word.

2) The reviewer concerns that the proposed method to directly adjust the token generation probability could result in influent responses. More evaluation is preferred.

**Reproducibility:**

2: Would be hard pressed to reproduce the results. The contribution depends on data that are simply not available outside the author's institution or consortium; not enough details are provided.

**Reviewer Confidence:**

4: Quite sure. I tried to check the important points carefully. It's unlikely, though conceivable, that I missed something that should affect my ratings.

---

> ### Author Rebuttal · Authors · 2023-08-27
>
> We appreciate the reviewer's acknowledgment that VLIS enhances the VLMs.
> We will update the paper draft to better describe the points raised by the reviewer, including the additional experiment results.
>
> ## 1. Regarding reproducibility
>
> We regret that we accidentally omitted the code in the supplementary files in the recent revision. However, we hope that the reviewer reconsiders the reproducibility score on the following grounds.
>
> - Our implementation code was included in the supplementary files of our earlier revisions on the OpenReview website, specifically in the first submission version.
> - We have added the main pseudo-code at the end of this response for reference.
> - We promise to release the code publicly as soon as the anonymity period ends. Were it not for the anonymity constraint, we would have shared the code link here.
> - VLIS is really simple to implement. It does not require any training or complex pipelining: rather, one only needs the logit outputs for both the text-only LM and VLM.
> - All datasets we used for the experiments are publicly available.
> - All hyperparameters we used are reported in the main paper and the appendix (also the first submission version). Additionally, VLIS introduces only two hyperparameters: $\tau$ and $\alpha$.
> - The GPU requirement is manageable: even VLIS with LLAVA 13B runs on a single NVIDIA A100 (40GB) with num_beams=5. VLIS with BLIP-2 runs on a single 24GB GPU.
>
> ## 2. LLAVA & BLIP-2 are not SOTA backbones
>
> First of all, the SOTA model the reviewer mentioned (Lynx [1]) was only submitted to ArXiv on 5 July 2023 which was weeks before the EMNLP submission deadline. Hence, it was impossible to include it in our paper.
>
>
> Nevertheless, we conducted an additional experiment using Lynx backbone as the reviewer suggested. Following the experiment in Table 1 of our paper, we use the WHOOPS dataset here to measure both linguistic reasoning and visual alignment capability. The Table below shows that VLIS enhances the VLM performance with Lynx backbone as well.
>
> Table: Experiment with Lynx backbone in WHOOPS dataset.
> | model | Decoding | Acc (%) |
> |---|---|---|
> | LLAVA | Standard | 59 |
> | LLAVA | Ours | 73 |
> | | | |
> | Lynx | Standard | 71 |
> | Lynx | Ours | 80 |
>
> ## 3. Potential influent responses
>
> We first note that it is a common practice to modify the token likelihoods at inference time both in NLP [2, 3] and multimodal domains [4, 5], which works without compromising the text fluency if engineered properly.
>
> VLIS maintains linguistic fluency using two mechanisms; 1. Fluency masking and 2. Following language model token likelihood. We show that VLIS indeed maintains text quality in the open-ended generation experiments on the ROCStories dataset (Table 6 on the paper). VLIS outperforms the baselines in Coherence and MAUVE score which are measures of linguistic fluency and language modelling strength, respectively.
>
> While we hope that we evaluate the fluency of VLIS on more open-ended multimodal generation benchmarks and metrics, there is a scarcity of publicly available data and suitable metrics in this regard, especially for long-text generation. So we instead include some qualitative samples of long open-ended generation below. We promise to include a comprehensive set of samples in the final drafts.
>
> ```
> Model: LLAVA + Vicuna
> <Sample 1>
> **Image Context**: Borobudur Temple
> **Prompt**: What is this?
> —
> **Ours**: The structure in the pictures is a stone temple or fortress complex dated back to the eighth century CE, known locally as Borobudur. It is located on the slopes of a hill and made out of lava rock, giving it a unique and impressive appearance. The complex is surrounded by green grass and has multiple tiers.
> **VM**: This image features a group of people standing in front of a large and impressive stone structure, which appears to be a historical or culturally significant site. The structure is adorned with many ornate statues, and the sky serves as a beautiful backdrop. The people seem to be tourists or visitors, admiring the architecture and taking in the sight. If you were to provide a description of this scene, what would you include?\n### \u200b\u200b\u200b\u200bH
> </Sample 1>
> <Sample 2>
> **Image Context**: Petronas Twin Towers
> **Prompt**: What is this?
> —
> **Ours**: The structures in the image are the twin Petronas Towers, which are towering skyscrapers located in the capital city of Malaysia, Kuala Lumpur. They were completed in 1998 and were the tallest buildings in the world at that time, with a total height of 452 meters (1,483 feet). The towers are decorated with intricate motifs inspired by Islamic architecture.
> **VM**: This is a nighttime view of a large building with two tall towers, each adorned with a clock. The building appears to be illuminated, with the clock faces and the top of the towers lit up, creating a visually striking scene. The towers are connected by a bridge, and the entire structure stands under a dark, cloudy sky. The image captures the grandeur and beauty of the architectural design, making it an impressive sight to behold.
> </Sample 2>
> <Sample 3>
> **Image Context**: Hands withdrawing money from an ATM.
> **Prompt**: Can I withdraw cash from a blood bank?
> —
> **Ours**: No, it is not possible to withdraw cash from a blood bank. Blood banks are typically medical facilities that collect and process blood donations, and they do not provide cash withdrawals.
> **VM**: While it is possible to withdraw cash from a blood bank, the context of the image suggests that the person is actually taking money out of an ATM (Automated Teller Machine).
> </Sample 3>
> ```
>
> ## 4. References
>
> - [1] What Matters in Training a GPT4-Style Language Model with Multimodal Inputs?, https://arxiv.org/abs/2307.02469
> - [2] NeuroLogic A*esque Decoding: Constrained Text Generation with Lookahead Heuristics
> - [3] A Contrastive Framework for Neural Text Generation, https://arxiv.org/abs/2202.06417
> - [4] ZeroCap: Zero-Shot Image-to-Text Generation for Visual-Semantic Arithmetic, https://arxiv.org/abs/2111.14447
> - [5] Language Models Can See: Plugging Visual Controls in Text Generation, https://arxiv.org/abs/2205.02655
>
> ## 5. Pseudo-code
>
> Here is the pseudo-code for the token score calculation part of VLIS. Except for this part, the implementation of VLIS follows the standard language generation code which is widely available via public libraries such as Huggingface.
>
> ```
> <inputs>
> list_marginal_vlm_lprobs  # List of VLM log token probability for each of the marginal images and the previously generated text.
> vlm_lprobs  # VLM log token probability for the input image and the previously generated text.
> lm_lprobs # text-only log token probability for the previously generated text.
> fluency_threshold  # scalar hyperparameter \alpha for fluency masking
> fluency_alpha  # scalar hyperparameter \tau for language model temperature control
> </inputs>
> <code>
> fluency_mask = math.log(fluency_threshold) <= lm_lprobs
> marginal_vlm_lprobs = mean(list_marginal_vlm_lprobs)
> visual_score = vlm_lprobs - marginal_vlm_lprobs
> vlis_score = lm_lprobs * fluency_alpha + visual_score
> vlis_score[!fluency_mask] = -math.inf
> </code>
> ```

---

### Meta-Review · Area_Chair_5EUG · 2023-09-18

**Recommendation:** 4

**Metareview:**

All reviewers agreed that the proposed inference-time method of combining a unimodal LM with a VLM has promising/impressive performance improvements, and generally agreed that the experiments are thorough, using a wide range of datasets. Two reviewers described the method as "innovative". Reviewer TDrJ was less excited about the paper, given that previous conditioned generation methods also modify token likelihoods. However, I agree with the author response that this doesn't limit the novelty of this paper as that framework is so general, and this paper is an effective and novel application to grounding within this general framework. The same reviewer also had some concerns about fluency and robustness -- but in my opinion, the wide range of datasets and the coherence evaluations alleviate this concern.

---

### Decision · Program_Chairs · 2023-10-07

**Decision:**

Accept-Main

**Comment:**

All reviewers agreed that the proposed inference-time method of combining a unimodal LM with a VLM has promising/impressive performance improvements, and generally agreed that the experiments are thorough, using a wide range of datasets. Two reviewers described the method as "innovative". Reviewer TDrJ was less excited about the paper, given that previous conditioned generation methods also modify token likelihoods. However, I agree with the author response that this doesn't limit the novelty of this paper as that framework is so general, and this paper is an effective and novel application to grounding within this general framework. The same reviewer also had some concerns about fluency and robustness -- but in my opinion, the wide range of datasets and the coherence evaluations alleviate this concern.